# A New Octagonal Close Ring Resonator Based Dumbbell-Shaped Tuning Fork Perfect Metamaterial Absorber for C- and Ku-Band Applications

**DOI:** 10.3390/mi13020162

**Published:** 2022-01-22

**Authors:** Md Salah Uddin Afsar, Mohammad Rashed Iqbal Faruque, Md Bellal Hossain, Air Mohammad Siddiky, Mayeen Uddin Khandaker, Amal Alqahtani, D. A. Bradley

**Affiliations:** 1Space Science Centre (ANGKASA), Institute of Climate Change (IPI), Universiti Kebangsaan Malaysia, Bangi 43600, Malaysia; P108627@siswa.ukm.edu.my (M.S.U.A.); p109529@siswa.ukm.edu.my (M.B.H.); armaneee037@gmail.com (A.M.S.); 2Centre for Applied Physics and Radiation Technologies, School of Engineering and Technology, Sunway University, Bandar Sunway, Selangor 47500, Malaysia; mayeenk@sunway.edu.my (M.U.K.); d.a.bradley@surrey.ac.uk (D.A.B.); 3Department of Basic Sciences, Deanship of Preparatory Year and Supporting Studies, Imam Abdulrahman Bin Faisal University, Dammam 34212, Saudi Arabia; amalqahtani@iau.edu.sa 15; 4Centre for Nuclear and Radiation Physics, Department of Physics, University of Surrey, Guildford GU2 7XH, UK

**Keywords:** perfect metamaterial absorber, octagonal close ring resonator, dual band applications, polarization insensitivity

## Abstract

In this paper, a new octagonal close ring resonator (OCRR)-based dumbbell-shaped tuning fork perfect metamaterial absorber for C- and Ku-band applications is presented. This design is a new combination of an octagonal ring close ring resonator with two dumbbell-shaped tuning forks metal strips integrated on epoxy resin dielectric substrate. The proposed perfect metamaterial absorber (PMA) is assessed by finite-integration technique (FIT)-based electromagnetic simulator-Computer simulation technology (CST) software. The anticipated assembly reveals dual resonance frequencies of 6.45 GHz and 14.89 GHz at 99.15% and 99.76% absorption, respectively, for TE incidence. The projected design is augmented through various types of parametric studies, such as design optimization, the effect of the octagonal ring resonator width, and varying the split gap of the double tuning fork. The numerical results are also investigated and verified using the equivalent circuit model, another electromagnetic simulator high frequency structural simulator (HFSS), and different array combinations that showed very negligible disparity. The TE polarization wave is applied to analyze the absorption separately and oblique incidence angle showing polarization insensitivity up to 30° and wide incident angle up to 60°. The presented metamaterial absorber is suitable for satellite communication bands, stealth-coating technology, and defense and security applications.

## 1. Introduction

Non-natural metamaterials (MMs) are engineered combinations consisting of sub-wavelength conducting material (metal) sandwiched with a dielectric medium. These sorts of designs provide some unusual properties which are not naturally present [1]. For these eccentric electromagnetic properties of plentiful metamaterials, namely negative permittivity (ENG), negative permeability (µ-negative), negative refractive index (NRI), and cloaking, the structure scheme and practical uses of metamaterials has become an important topic in recent research [2]. The principle characteristic of a perfect EM absorber is that it will not reflect or transmit the incident electromagnetic wave. An absorber highlights the maximizing of losses of scattering of EM fields including reflectivity and transmissivity at expected frequencies.

The first metamaterial perfect absorber (MPA) involves three deposits containing two metallic layers and a substrate of dielectric, and exhibits a simulated absorptivity of nearly 99% at 11.48 GHz [3]. Electromagnetic radiation absorber can be classified into two categories- resonant absorber and broadband absorber [4]. When an incidental EM wave interacts with the materials in a resonant setup at a specific frequency it is called a resonant absorber; where the wavelength related to the resonant frequency $(\omega_{0}\;);{\;\lambda }_{0}=\frac{2\pi c}{\omega_{0}}$, and where c is the speed of light. On the other hand, broadband absorbers usually depend on materials whose characteristics are non-dispersive and therefore can absorb radiations over a bulky bandwidth [5]. Consequently, the absorber materials mostly absorb the power of the impacting wave. The enactment of an absorber is influenced by its depth and structural morphology and also the constituents used to manufacture it [6]. Recently, most of the innovative works on MAs have put emphasis on single band and dual band perfect metamaterial absorber (PMA) whereas there are only a few research studies on multi-band perfect metamaterial absorbers [7,8]. In fact, in refs. [9,10] a structure containing two or more planar resonant unit were designed and investigated but they did not function for dynamic adjustment. Landy et al. in 2008 designated several metamaterial absorbers that received interest from scientists with perfect absorbers in the millimeter to nanometer wavelength ranges [6], a polarization-insensitive absorber [11], a broadband absorber [12], a multiband absorber, ultra-broadband, and a refractive index sensor.

Electric and magnetic resonance occurring in a perfect metamaterial absorber (PMA) are able to unconventionally adapt the effective permittivity and permeability. The perfect metamaterial absorbers have some special advantages compared to the outmoded absorbers including the Salisbury screen [13], ferrite, and wedge tapering [14]. In addition, a thin-film substrate which may attain an atypical condition of absorbance. Furthermore, a highly efficient absorber can be comprehended using tunable gadgets or materials. These functions of metamaterial absorbers make them novel and make them fit to be candidates for different uses in the millimeter wavelength range of the optical band’s spectra for example, photodetectors, solar photovoltaic, and thermo-photovoltaic bands [15,16,17]. PMAs functioned at the resonant frequency and reveal small absorption due to their intermittent clusters of resonators.

Several prior studies on high-absorptivity perfect metamaterial absorbers were designed to describe their absorbance features [18]. For instance, the polarization-independent PMAs conveyed by Kollatou et al. revealed better performance of absorption properties in the microwave range [19]. An absorber called the Jerusalem Cross of dimensions 24 × 24 mm^2^ describes absorption above 95% and which resonantes at 14.75 GHz and 16.1 GHz, respectively [20]. Lin et al. suggested a metamaterial absorber unit cell assembly with magnitudes 10.92 × 10 mm^2^ that was used in a microwave system that showed the highest absorbance 99.6% at a resonance frequency of 2.38 GHz [21]. To analysis the absorption properties, Islam et al. designed a multi-band split S-shaped metamaterial absorber which absorbed a maximum of 55% of incident electromagnetic radiation [22]. A broadband perfect metamaterial absorber of dimension 10 × 10 mm^2^ on the basis of electric split ring resonator (ESRR) loaded with lumped resistors presented by Zhao et al. It showed the absorption rates of 99.3%, 97.1% and 98.6% at the resonance frequencies 5.45 GHz, 15.46 GHz, and 19.48 GHz, respectively [23].

In 2015, Dincer et al. [24] recommended a design scheme of absorbing metamaterial which absorbed nearly full incident EM radiation on it. It was measured, characterized, and finally recognized as a perfect metamaterial absorber (PMA) with an outstanding absorbance of 99.99% at 5.48 GHz and 99.92% at 0.865 THz. A terahertz dual band MA was proposed by Wen et al. that showed two distinct absorptions of nearly 81% and 63% at 0.45 THz and 0.92 THz, respectively [25]. A multilayer dual band metamaterial absorber was presented by Kim et al. in megahertz area [26]. It showed average absorptions of 96% in the 4 to 6 GHz frequency range. Faraji (2015) proposed a variable terahertz absorber [27]. Its characteristics were analyzed numerically, and it absorbs 96–99% at the resonance frequency range of 2.14–9.94 GHz. A Jerusalem cross metamaterial of dimension 24 × 24 mm^2^ with roamed load absorber represents absorptions of more than 95% at resonance frequencies of 14.75 GHz and 16.1 GHz [28]. Montaser et al. [29] designed a new metamaterial-based absorber for multiple applications at Gigahertz and Terahertz frequencies. In the numerical simulation at the resonance frequencies of 2.8 GHz, 4.1 GHz and 5.8 GHz, it showed absorbance of 99%, 99% and 89%, respectively. It exhibits maximum absorbance of 99.46% at 265.8 THz and 99.40% at 556.4 THz. A cross shape design of metamaterial absorber in 2017 was presented by Bağmancı et al. [30] with a size of 500 × 500 nm^2^. It absorbed a high rate of absorbance, 98.40% of incidental EM waves. A triple hexagonal SRR-based metamaterial sensor was reported in [31] to detect fuel (petrol, kerosene, and grease) contaminants which are used for real time applications. Moreover, Y. I. Abdulkarim et al. developed a broadband metamaterial absorber which is based on square shaped water metamaterial [32]. A newly improved version of C shaped SRR metamaterial with artificial neural network was designed to show improved quality factors because of using ANN on the design and determine dimension of the MTM structure [33].

In this paper, a dual band (C- and Ku-) octagonal close ring resonator (OCRR)-based dumbbell-shaped tuning fork metamaterial absorber is presented. It exhibits dual resonance (C, Ku) at 6.45 GHz and 14.89 GHz with an absorption rate of 99.15% and 99.76% respectively. Additionally, an attempt has been made to investigate the performance of the proposed PMAs by alternating the polarization angle and oblique incidence angle in the transverse electric (TE) mode of unit cell, array combination, and different widths of octagonal ring resonator and split gap of double tuning fork. The dimension of proposed unit cell is 8 × 8 mm^2^ which is very compact compared to the PMAs as stated in [19,22,34,35,36,37]. Moreover, the proposed design is very simple and highly symmetric structure that can be operated easily. The pluses of the designed structure are FR-4 substrate and insensitive to polarization angle up to 30°.

## 2. Design of the Unit Cell and Simulation

Two tuning forks with a dumbbell shape dual band (C- and Ku-) metamaterial absorber structure based on octagonal close ring resonator (OCRR) are shown in Figure 1. FR4 of thickness 1.6 mm is used as substrate material with dielectric constant of 4.6 and loss tangent of 0.025. The parameter of the unit cell is shown in Table 1. The dimension of the FR-4 substrate is 8 × 8 × 1.6 mm^3^. The schematic diagram of the proposed PMA unit cell design comprises a dielectric substrate, which is crammed in with two metallic layers which are shown in Figure 1a. The perspective view of the unit cell is shown in Figure 1b. The entire upper metallic layers including OCRR are made of copper which has a conductivity of 5.8 × 10^7^ Sm^−1^. The octagonal ring resonator is attached with two tuning forks and a dumbbell-shaped copper (pure) strip is placed between two tuning forks. The bottom layer is a thin copper sheet of dimension 8 × 8 × 0.035 mm^3^, as shown in Figure 1c. The radius of the bigger outsider octagon and smaller inner octagon are 4 mm and 3.5 mm, respectively. Therefore, the width of the each OCRR is 0.50 mm. The gap (G) between two tuning forks is 0.40 mm. The length of the handle (a), breadth (c) and height (d) of the tuning forks are 1.8 mm, 4 mm and 1.3 mm respectively. The size of each head of the dumbbell is 1 × 1.6 mm^2^ and the dumbbell handle is 0.5 × 1 mm^2^.

A FIT-based CST simulator is familiarized to simulate the projected design for comprehensive study of the characteristics structured in this article. The incidence wave is applied along the *Z*-axis while the unit cell boundary conditions are applied along *X*-axis and *Y*-axis. *Z*-axis is also used for open space condition.

## 3. Method and Physical Explanation

Using the following equation, the absorbance efficiency can be calculated-
(1)$$A\left(\omega \right)=1-{\left|S_{11}\right|}^{2}-{\left|S_{21}\right|}^{2}=1-\left[R\left(\omega \right)+T\left(\omega \right)\right]$$
where A(ω), R(ω) and T(ω) denote the efficiency of absorption, reflection and transmission respectively [38,39,40]. If *R*(*ω*) and *T*(*ω*) are at a minimum then absorption *A*(*ω*) will be at the maximum. As the copper back is enough thick to inhibit all incident EM wave i.e., *T*(*ω*) = 0 then the absorption efficiency, *A*(*ω*) = *A*(*ω*)_max._ In this case, reflection efficiency is the only consideration to obtain the maximum absorption which follows the equation
(2)$$A\left(\omega \right)=1-{\left|S_{11}\right|}^{2}=1-R\left(\omega \right)$$

To achieve the expected dual band resonance from a metamaterial, the physical unit cell parameters should be cautiously optimized. The simulated result of reflection coefficient (S11), transmission coefficient (S21), and absorption scenario is presented in
Figure 2. For a better understanding of the physical sensitivity and performance of the absorbers, the electric and magnetic field and the surface current distribution are investigated in addition to the surface current for all resonance frequencies. 

A wide-ranging physical analysis is accomplished in this segment to comprehend the authentic activities of this metamaterial absorbers. First, we concentrated on the query as to how every sub-module keeps its own behavior in such as a quarantined device. As a basic physical elucidation, the connection with the electrical permittivity (ε*_r_*) and the electric polarization (*P*) can be defined by the following fundamental equation
(3)$$D=\in_{0}E+P_{e}=\left(1+\chi_{e}\right)\in_{0}E\;\mathrm{or},\;\frac{D}{E}=\in_{0}\left(1+\chi_{e}\right)\ldots$$

With the condition $\chi_{e}$ < −1, then ε*_r_* turns into the adverse, fulfilling the identical sub wavelength situation. The similar tactic may be applied to scrutinize the association with the effective magnetic susceptibility$\;(\chi_{m}$) and the magnetic field (*B*)
with the second corresponding formula
(4)$$\;\;B=\mu_{0}H+\mu_{0}P_{m}=\left(1+\chi_{m}\right)\mu_{0}H\;\mathrm{or},\;\frac{B}{H}={\;\mu }_{0}\left(1+\chi_{m}\right)$$

Finally, the metamaterial perfect absorber (PMA) efficiency is studied here to deliberate accurate losses in the metals, which are called surface power loss and volume losses in the dielectric substrates called volume loss power [41]. In this regard, the simulation and the power loss estimation have been investigated with a FIT-based CST microwave studio. The dissolute power on the imperfect dielectric layer roughly depends on the parameters of the loss tangent value taken as *tanδ* = 0.005.

## 4. Results and Discussion

Updated microwave simulator CST software was activated to simulate the whole spectrum. Absorption, reflection, and transmission spectra for the suggested perfect metamaterial absorber (PMA) are depicted in Figure 2. At normal incidence, the simulation results show higher absorbance at all dual resonance frequencies which are 99.15% and 99.76% at 6.45 GHz and 14.89 GHz respectively that remains within the C and Ku bands with remarkable bandwidth. The transmission coefficient is zero for all resonance frequencies because of keeping the completely metallic thin shift at the bottom of the unit which is equal to the dimension of the substrate. The following equation is used to calculate the average absorption efficiency of the projected metamaterial:
(5)$$A_{avg}=\frac{1}{\lambda U-\lambda L}\int_{\lambda L}^{\lambda U}A\;\left(\lambda \right)d\lambda \;$$here, *A*(λ) is the wavelength response of the absorption coefficient.

Polarization insensitivity and independency of the oblique incidence angle are the important tasks of the perfect metamaterial absorber design. Both of these properties play the vital role of overcoming the typical drawback of an MA with a single polarization and a small working angle. An ideal absorber is one that operates the polarization angle independently as well as having a wide working angle, and is capable of absorbing approximately unity of the projected EM radiation that happen at normal angles. The simulated result of the amplitude of absorption of the proposed PMA for TE polarization incident waves is shown in Figure 3. At 0°, the absorption at the dual resonance frequencies 6.45 GHz and 14.89 GHz is 99.15% and 99.76%, respectively. If the polarization angle changes to 30°, the resonance frequencies change slightly to a higher frequency. The two resonance frequencies were changed to 6.816 GHz and 16.128 GHz with 94.5% and 98.36% absorption, respectively. Then the polarization angle changes to 60°. Therefore, the corresponding resonances are observed at 6 GHz, 11.87 GHz, and 13.5 GHz with 47.9%, 97.17%, and 89.5% absorption, respectively. For 90°, the resonance frequencies are 11.9 GHz, and 13.5 GHz with 73%, and 71.5% absorption, respectively. It is mentioned that the proposed absorber shows the polarization angle unaffected up to 30° and showing polarization sensitivity up to 90°. The amplitude of absorption for different oblique incidence angles, such as θ = 0°,30°, 60° and 90° are analyzed in Figure 4. It is seen in Figure 4 that as the oblique incidence angle increases, the absorption frequency of the absorber changes slightly to a lower frequency with high absorptivity. However, as the angle of oblique incidence changes, the absorption will of course decrease as the capability of the MMs’ structure to conduct the flowing current in the dielectric between the two metal layers is reduced. Therefore, the proposed absorber is wide, with an oblique incidence angle up to 60°. Effective permittivity and permeability are two important factors to explain the properties of metamaterials. These parameters can also be obtained using the scattering parameter. The effective permittivity and permeability are shown in Figure 5a,b. As can be seen from this figure, both effective permittivity and permeability have negative values around the resonance frequency (ε_eff_ < 0 and µ_eff_ < 0). The absorption technique of the offered MA could be well explained by investigating the electric field distribution, magnetic field distribution and surface current density. These three phenomena are depicted in Figure 6, Figure 7 and Figure 8 at the two resonance frequencies 6.45 GHz and 14.89 GHz, respectively. The induced electric field, magnetic field, and surface current are strongly centered on the octagonal ring as well as on the metal strip (dumbbell-shaped tuning fork) of the unit cell. With the increase of the frequency, the electromagnetic resonance shifted to the top layer of the unit cell. It is observed that the octagonal ring is sensible for the absorption peak at 6.45 GHz while absorption peak at 14.89 GHz is seen due to tuning fork along with dumbbell shape. It is seen no doubtfully, the surface electric flow is mostly centralized along the octagonal ring shape and tuning fork resonator, creating a strong magnetic field. In addition, the induced E-field is opposite to the H-field. Therefore, the design simulates an equivalent LC architecture, the resonance of which can be adjusted using control parameters.

Finally, a comparison between the performance of the proposed PMA and different published PMA is shown in Table 2. Some factors, for example absorption rate, absorption band application, metamaterial structure, and dimension have been inspected.

## 5. Parametric Study of the PMA Unit Cell

### 5.1. Design Optimization

The suggested PMA development was settled on the basis of the trial-and-error technique to acquire the maximum resonance frequencies with excellent absorptivity. The design of the PMA structure was started by taking an octagonal close ring shape resonator that exhibits single resonance frequency (14.6 GHz) with 57.7% absorption, as indicated in Figure 9a. In the next step, two opposite tuning forks were added to the octagonal ring shape resonator as shown in Figure 9b. According to Figure 9b, this design shows two resonance frequencies (6.69 GHz and 15 GHz) with 97.7% and 99.7% absorption. The scheme was finalized by inserting a dumbbell shape resonator between the two tuning forks shown in Figure 9c. The final complete unit cell represented two main resonance frequencies (6.45 GHz and 14.89 GHz) with excellent absorption (99.15% and 99.76%). The absorption graph of all design is shown in Figure 9d.

### 5.2. Effect of Changing the Octagonal Ring Width

Moreover, the effect of changing the octagonal width on absorption is presented in Figure 10a, where both resonance frequencies shift to higher as octagonal width increases. As octagonal width increases, the equivalent inductance increases, causing the lower resonance frequencies swings to higher and upper resonance remain unchanged.

### 5.3. Effect of Varying the Tuning Fork Split Gap

The correlation with the opening end width of the double tuning fork and simulated result scenario is indicated in
Figure 10b. With the increase or decrease of the split gap, the net capacitance changes inversely. Thus, the minor resonance crests move to higher frequencies. Nevertheless, the greater absorption peak remains almost unbothered, which provides an easy approach to adjust distinct absorption frequencies.

## 6. Equivalent Circuit of the Proposed Metamaterial Absorber Unit Cell

The projected metamaterial absorber consists of both inductive and capacitive constituents. The metal bars of the of the octagonal close ring resonator (OCRR) and other metal strips involved in the unit cell acted as inductors, whereas the opening ends within the metal bar created capacitors. Hence, an *LC* resonance circuit is formed in this perfect metamaterial absorber (PMA) unit cell. Therefore, the resonance frequency (*n*) can be extracted from Equation (6) [42].
(6)$$2\pi n\sqrt{LC}=1\;\mathrm{or},\;n=\frac{1}{2\pi \surd LC}$$
where *C* and *L* are the capacitance and inductance of the capacitors and the inductors respectively. The capacitance formed by the splits can be derived by Equation (7)
(7)$$\frac{C}{\in_{r}}=\in_{0}\frac{A}{d}\left(F\right)$$
where $\in_{0\;}\mathrm{and}\;\in_{r}$ are the permittivity in free space and the relative permittivity respectively and *A* is the cross-sectional area of the conducting strip, *d* is the split gap.
(8)$$L\left(nH\right)=2\times 10^{-4}l\left[\ln\left(\frac{l}{w+t}\right)+1.193+0.02235\left(\frac{w+t}{l}\right)\right]C_{g}$$

Here, *Cg* = Correction factor, *w* = width, *l* = length and *t* = thickness of the strip.

Figure 11a denotes a predictable equivalent circuit for the considered metamaterial absorber unit cell. *L*1, *L*2, *L*3, *L*4, *L*5, and *L*6 of the circuit model refer to the inductors, while *C*1, *C*2, *C*3, and *C*4 represent the capacitors. The circuit was boosted and simulated using a simulator advanced design system (ADS). As such, inductor *L*1 and *L*2 are used for copper back and *C*1 is the coupling capacitor. The capacitors *C*2 and *C*3 are used for the gaps between two tuning forks which are responsible for the first resonance (6.45 GHz). While *C*4 is the coupling capacitor, *L*5 and *L*6 refer to two parallel inductors replacing the dumbbell shape metal strip, which enhances the second resonance (14.89 GHz).
Figure 11b represents a comparative assessment of the CST-simulated S11 and Advanced design software (ADS) that describes the nearby resemblances between these two S11 values.

## 7. Array Simulation of the Unit Cell

To make the results more acceptable, an array of different combinations of the proposed unit cell are simulated by CST. Surprisingly, the simulated results are closely similar to the other validations. The simulated results for different combinations of array versus unit cell are depicted in Figure 12a,b. They exhibit two resonances of average frequency 6.39 GHz and 14.90 GHz at −20.88 dB and −26.43 dB, respectively. The average rate of absorbance (shown in Figure 11b) for the respective frequency is 99.20% and 99.75%, which shows very negligible amount variation.

## 8. Validation Using HFSS

A finite element method (FEM)-based high frequency structure simulator (HFSS) is used to simulate the PMA unit cell to justify the CST result of absorbance. The HFSS result shows good agreement with the CST result shown as in Figure 13. It also exhibits the dual band resonance with a little change of frequencies with nearly same absorbance.

## 9. Conclusions

In conclusion, a dual band (C- and Ku-) perfect metamaterial absorber (PMA) based on a dumbbell-shaped tuning fork copper strip with octagonal closed ring resonator (OCRR) is proposed and investigated by numerical simulation (CST and HFSS) and equivalent circuit model in the 4 to 18 GHz frequency range. The designed absorber attains absorbance of over 99% in a wide bandwidth of a wave’s incident at angle 0°. The resonance peaks located at 6.45 GHz and 14.89 GHz with an absorption of efficiency 99.15% and 99.76%, respectively, in TE polarization mode. The proposed dual band absorber exhibits high absorption of up to 30° polarization insensitivity and wide oblique incidence angle of up to 60°. In addition, the simulated results for different combinations of the array (2 × 2, 4 × 4, 8 × 8) exhibit the same resonance with minor variations. Compared to the contemporary published absorber, it has the benefit of simple design and easy implementation. This work offers an exotic dual band metamaterial perfect absorber, which has a widespread application in satellite communication bands, defense, security, and stealth-coating technology. This exclusive metamaterial absorber design reformed some authentication progressions that led to its uniqueness.

## Figures and Tables

**Figure 1 micromachines-13-00162-f001:**
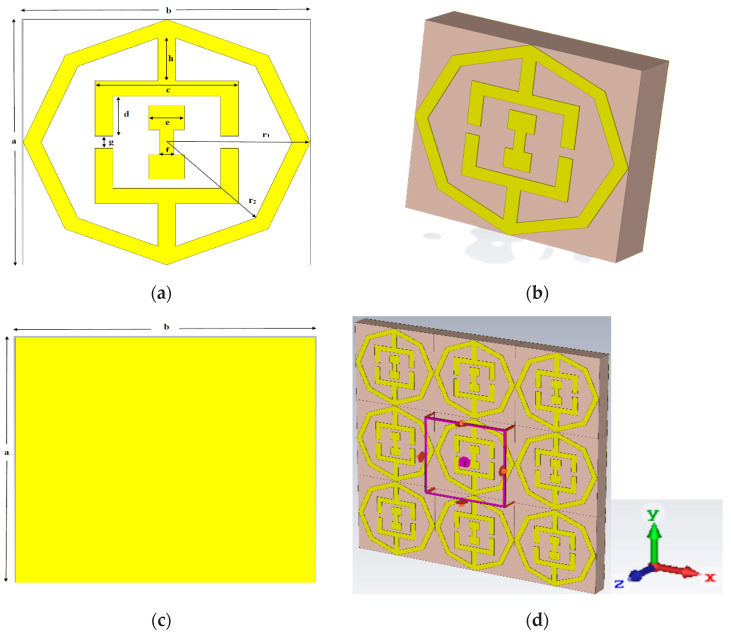
PMA unit cell (**a**) Front view (**b**) Perspective view (**c**) Back Copper (**d**) Boundary condition.

**Figure 2 micromachines-13-00162-f002:**
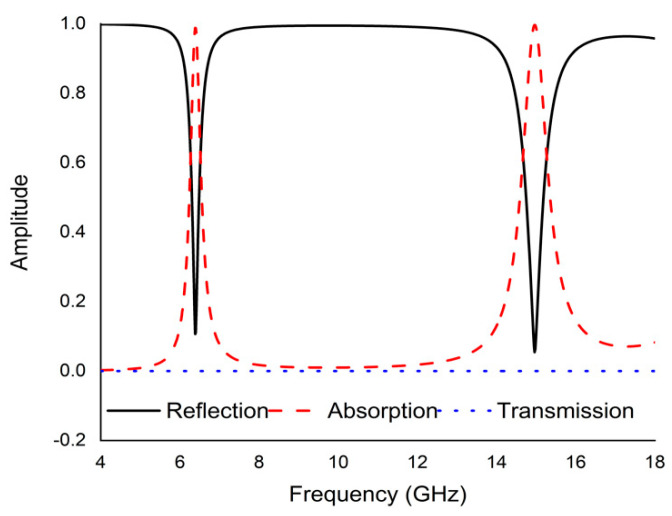
Simulated results for reflection coefficient, transmission coefficient and absorption condition.

**Figure 3 micromachines-13-00162-f003:**
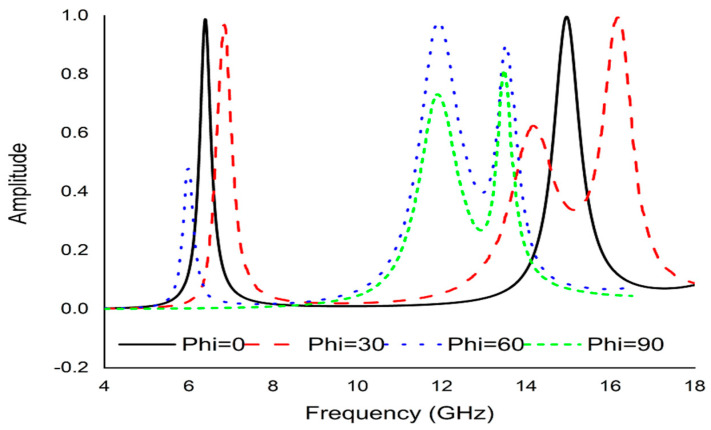
Absorption graph at different polarization angle.

**Figure 4 micromachines-13-00162-f004:**
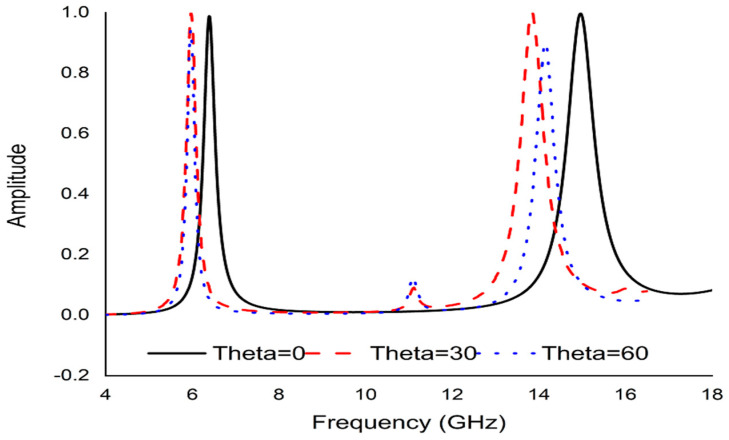
Absorption graph at different incident angle.

**Figure 5 micromachines-13-00162-f005:**
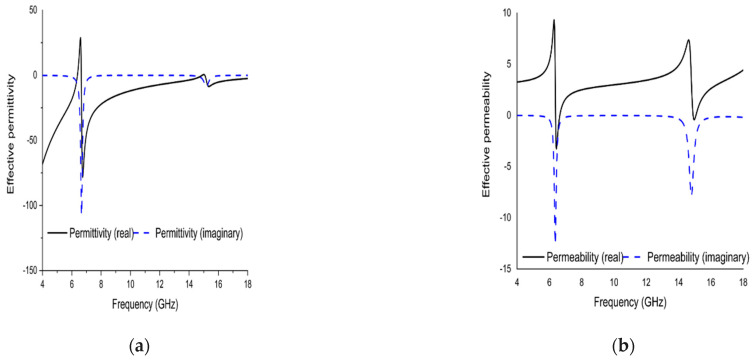
Graph at (**a**) effective permittivity (**b**) effective permeability.

**Figure 6 micromachines-13-00162-f006:**
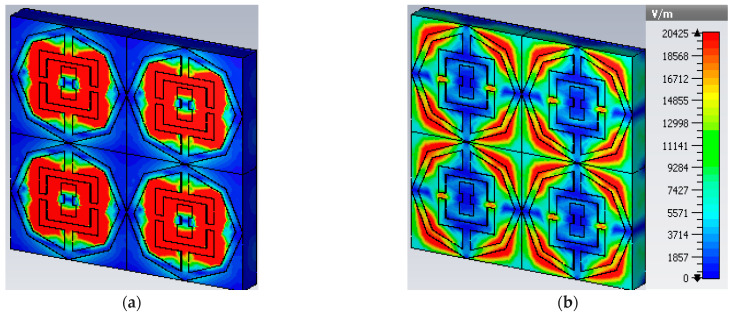
Electric field distribution (**a**) 6.45 GHz (**b**) 14.89 GHz.

**Figure 7 micromachines-13-00162-f007:**
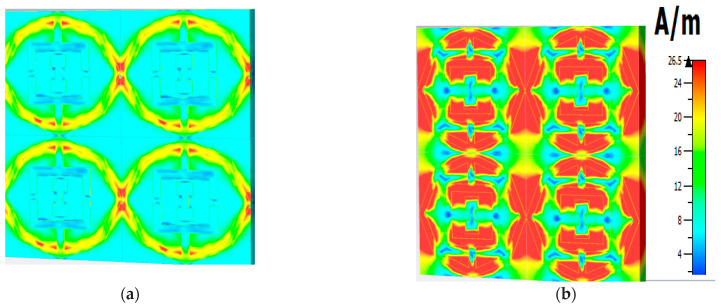
Magnetic field distribution (**a**) 6.45 GHz (**b**) 14.89 GHz.

**Figure 8 micromachines-13-00162-f008:**
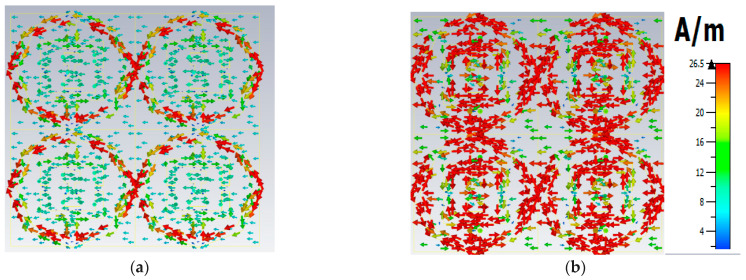
Surface current distribution (**a**) 6.45 GHz (**b**) 14.89 GHz.

**Figure 9 micromachines-13-00162-f009:**
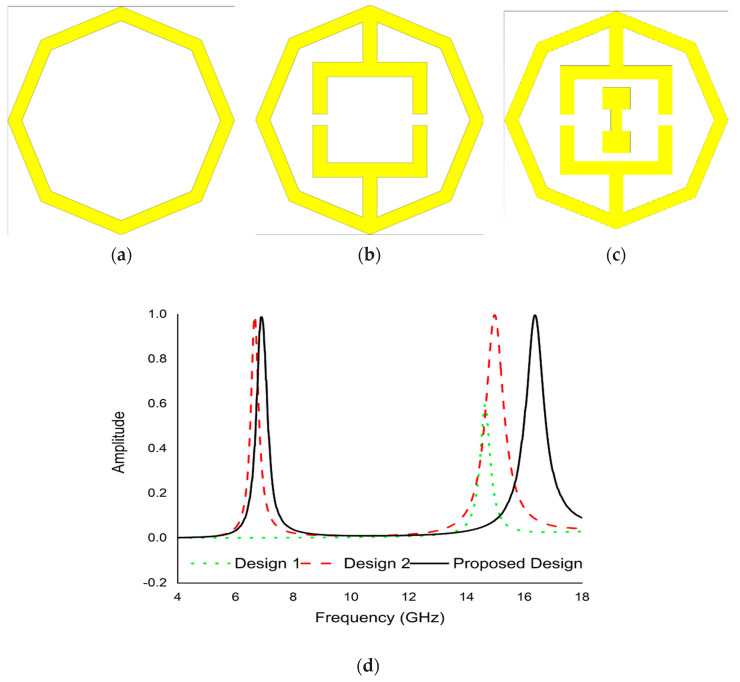
Design optimization (**a**) Design 1 (**b**) Design 2 (**c**) Design 3 (**d**) Absorption graph scenario for different design.

**Figure 10 micromachines-13-00162-f010:**
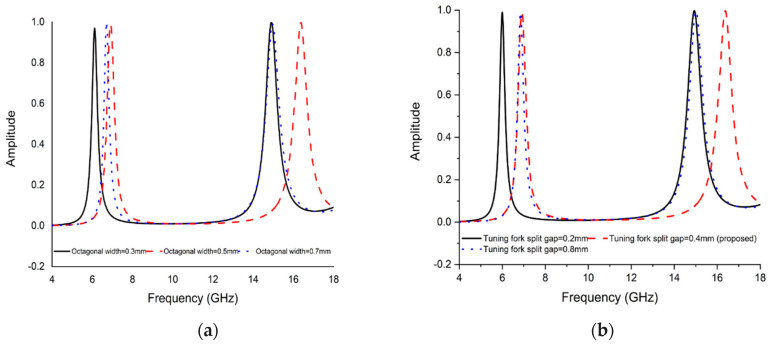
Absorption at (**a**) varying octagonal width (**b**) varying tuning fork split gap.

**Figure 11 micromachines-13-00162-f011:**
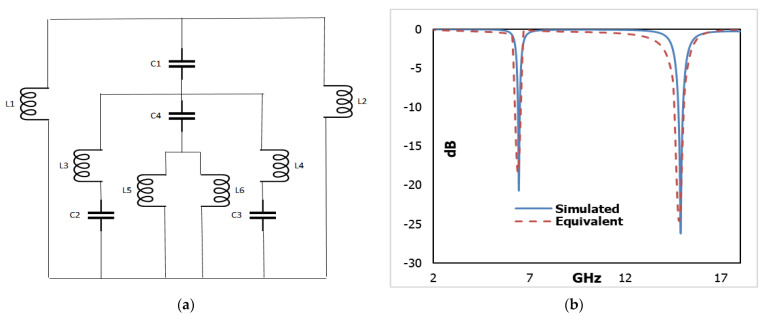
(**a**) Equivalent circuit model of unit cell (**b**) CST and ADS results of unit cell.

**Figure 12 micromachines-13-00162-f012:**
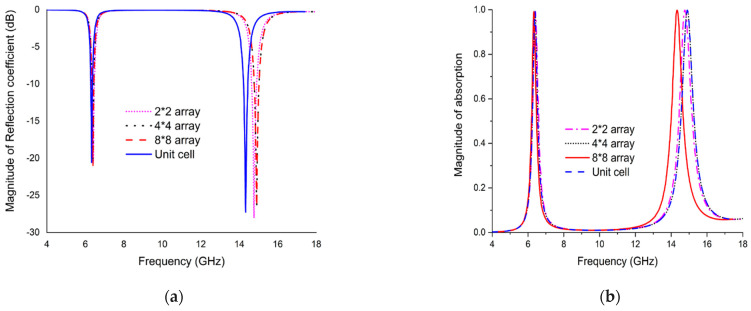
Comparison of CST results (**a**) S11 (**b**) Absorption.

**Figure 13 micromachines-13-00162-f013:**
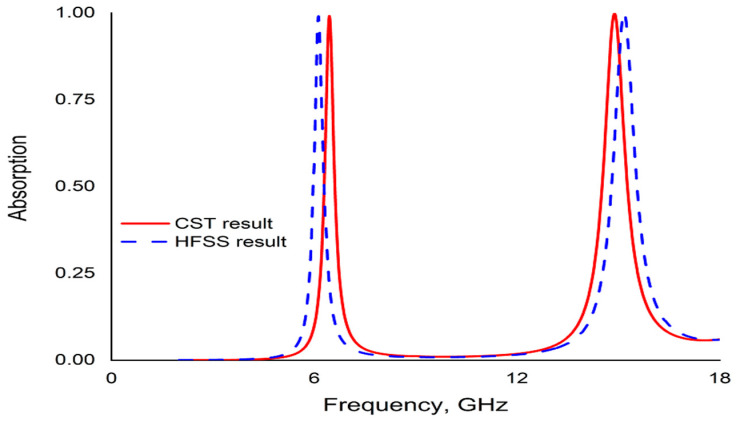
Comparison of absorbance by CST and HFSS result.

**Table 1 micromachines-13-00162-t001:** Parametric description of the unit cell.

Parameter	Dimension (mm)	Parameter	Dimension (mm)
a	8	c	4
b	8	d	1.3
h	1.8	e	1.6
g	0.4	f	1
r_1_	4	r_2_	3.5

**Table 2 micromachines-13-00162-t002:** Comparison of different published PMA with proposed PMA.

Author Name	Design Structure	Size (mm^2^)	Proposed Band	Absorption (%)	Year of Published
Proposed PMA	Tuning fork shape	8 × 8	C-Ku	99.76	--
M. F Zafar et a [37]	L-shape	8 × 8	X-band	90	2021
Islam et al [22]	S-shape	20 × 20	S-, X-, Ku-band	55	2017
Sen et al [36]	L-shape	9 × 9	X-, Ku-band	95	2017
M. M. Hasan et al [35]	Square shape	10 × 10	C-, X-, Ku-band	93	2017
Borah et al [34]	O-shape	12 × 12	X-band	98.90	2016
Kollatou et al [19]	Modified square	8 × 8	X-band	95.81	2013

## Data Availability

All the data is available within the manuscript.

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
