# Peer review of "A New Octagonal Close Ring Resonator Based Dumbbell-Shaped Tuning Fork Perfect Metamaterial Absorber for C- and Ku-Band Applications"

_micromachines, 2022, doi:10.3390/mi13020162_

Round 1

Reviewer 1 Report

Manuscript: A New Octagonal Close Ring Resonator Based Tuning Fork Dumbbell Shaped Perfect Metamaterial Absorber for Dual Band Applications

ID: 1559281

In this manuscript, authors suggest a dual band perfect metamaterial absorber structure which is tuning fork dumbbell shaped combination of octagonal close ring resonator. Proposed absorber structure has nearly unity absorption peaks at 6.45GHz and 14.89GHz. The present form of the submitted manuscript needs a major revision before the final review as followings;

  1. The title of the main manuscript is suitable but it is best to give resonance frequencies of bands in the title.
  2. The abstract section gives a brief information about the study, it is in a good form.
  3. Introduction section gives a comprehensive literature review by mostly recently published research articles, it is in a good form. However, the following studies could improve the introduction section;
  • doi.org/10.1007/s10854-021-06891-6
  • doi.org/10.1016/j.jmrt.2021.05.031
  • doi.org/10.1117/1.OE.60.4.047106
  1. Please explain well, how to obtain design parameters. It is critical in the design steps.
  2. Please explain the advantages of the octagonal shape compared with the circular shaped resonator.
  3. Also explain the metamaterial behaviour of the proposed absorber structure, why the title “metamaterial” is used for this designed resonator?
  4. Please explain the aim of using fork dumbbell resonator in the octagonal ring resonator, it is not clearly explained. Advantages must be given clearly.
  5. Please explain the boundary conditions in the simulation setup.
  6. Please explain why TE mode absorption is proposed, is it enough for applications?
  7. Proposed design is not polarization angle and incident angle independent under TE polarization, is it an advantage or not? Please explain.
  8. The conclusion section is in a good form.

Author Response

As attached.

Reviewer 2 Report

The authors present an interesting simulation work on a fork-dumbbell shaped perfect metamaterial absorber, revealing dual band resonance and high absorption. The authors provide details on simulation design, physics explanation, design optimization and tuning, and validate their simulations by comparing CST and HFSS results. I have some questions and comments before sending for publication at Micromachines.

What is the novelty or difference compare to the author's previous work Micromachines 2021, 12(8), 878?

In the introduction, 1) the authors provide one single paragraph with too many details stacked, I recommend to reorganize and make the structure more coherent and concise. 2) By presenting multiple examples on recent PMA designs, the authors should provide some discussions on what is lacking, or what are the existing challenges, how their design would address the current technical or scientific problems. 

Page 10, Table 2, please add references at each presented example in the table.

What would be possible limiting factors for discrepancies between real device fabrications compared to the simulation? How to address such discrepancies to make real perfect metamaterial absorber? Please explain.

And, please check grammar errors of the manuscript. 

Author Response

As attached.
